# Minimum dietary diversity and its determinants among lactating mothers in five Sub-Saharan African countries: A multilevel analysis

**Alebachew Ferede Zegeye**[1]*, **Tadesse Tarik Tamir**[2], **Desale Bihonegn Asmamaw**[3], **Desalegn Anmut Bitew**[3], **Elsa Awoke Fentie**[3], **Bewuketu Terefe**[4], **Rahel Mulatie Anteneh**[5], **Lemlem Daniel Baffa**[6], **Misganaw Guadie Tiruneh**[7], **Kaleb Assegid Demissie**[7], **Tadele Biresaw Belachew**[7], **Wubshet D. Negash**[7,8], **Melak Jejaw**[7]

1 Department of Medical Nursing, School of Nursing, College of Medicine and Health Sciences, University of Gondar, Gondar, Ethiopia, 2 Department of Pediatric and Child Health Nursing, School of Nursing, College of Medicine and Health Sciences, University of Gondar, Gondar, Ethiopia, 3 Department of Reproductive Health, Institute of Public Health, College of Medicine and Health Sciences, University of Gondar, Gondar, Ethiopia, 4 Department of Community Health Nursing, School of Nursing, College of Medicine and Health Sciences, University of Gondar, Gondar, Ethiopia, 5 Department of Public Health, College of health science, Debre Tabor University, Debre Tabor, Ethiopia, 6 Department of Human Nutrition, Institute of Public Health, College of Medicine and Health sciences, University of Gondar, Gondar, Ethiopia, 7 Department of Health Systems and Policy, Institute of Public Health, College of Medicine and Health Sciences, University of Gondar, Gondar, Ethiopia, 8 National Centre for Epidemiology and Population Health, The Australian National University, Canberra, Australia

* alexferede24@gmail.com

## Abstract

### Background

Ensuring minimum dietary diversity is crucial for lactating mothers. It allows them to consume a variety of foods from different groups, meeting their nutritional needs and supporting maternal and infant health. Despite the global impact of undernutrition and micronutrient deficiencies, the prevalence and determinants of minimum dietary diversity during lactation remain poorly recognized in developing countries. This research aims to assess minimum dietary diversity and its determinants among lactating mothers in five Sub-Saharan African countries.

### Methods

Data from the most recent Demographic and Health Surveys, which covered five Sub-Saharan African countries between 2021 and 2023, were used to execute secondary data analysis. This study included 19,917 lactating mothers in total. Using a multilevel mixed-effects logistic regression model, the variables associated to the minimum dietary diversity were established. Significant factors associated with the minimum dietary diversity were found at p-values <0.05. The adjusted odds ratio and 95% CI were used to interpret the result. The model with the highest loglikelihood ratio and lowest deviance was declared to be the best fit.

**Data availability statement:** In this study, we utilized the most recent data from the Burkina Faso, Kenya, Ghana, Mozambique, and Tanzania Demographic and Health Surveys, which is publicly available at the DHS Program website and is publicly available online at https://www.dhsprogram.com. Additionally, supplementary datasets employed for analysis will be uploaded alongside the manuscript submission to ensure transparency, verification, and replication purposes up on request.

**Funding:** The author(s) received no specific funding for this work.

**Competing interests:** The authors declared that there is no competing interest.

## Results

The magnitude of minimum dietary diversity among lactating mothers in Burkina Faso, Ghana, Kenya, Mozambique, and Tanzania was 25.66% (95% CI: 24.47, 25.75). Factors such as secondary and above educational level (AOR = 1.38, 95% CI: 1.18, 1.61), employed mothers (AOR = 1.40, 95% CI: 1.26, 1.56), distance which was not a big problem to access health facilities (AOR = 1.35, 95% CI: 1.21, 1.51), health facility delivery (AOR = 1.25, 95% CI: 1.08, 1.45), rich wealth status (AOR = 1.86, 95% CI: 1.60, 2.17), high community ANC utilization (AOR = 1.18, 95% CI: 1.04, 1.35), and reside in Ghana (AOR = 4.21, 95% CI: 3.60, 4.94) had higher odds of minimum dietary diversity.

## Conclusions

This study reveals that lactating mothers have low dietary diversity. Both community-level and individual-level factors impact this diversity. Consequently, health ministries in Burkina Faso, Kenya, Ghana, Mozambique, and Tanzania should prioritize women who underutilize antenatal services and those without formal education when designing strategies and policies.

## Background

Dietary Diversity (DD) among lactating mothers is the practice of consuming a wide range of foods across different food groups to meet the nutritional needs of their health and that of their breastfeeding child [1,2]. Minimum Dietary Diversity (MDD) among lactating mothers is the intake of variety food groups within a 24-hour period, which is essential for providing the necessary nutrients to support the health of both the mother and the breastfeeding infant [3–5].Globally, undernutrition and micronutrient deficiencies impact nearly half the population, particularly affecting lactating mothers. These deficiencies, especially in iron, iodine, vitamin A, and zinc, lead to serious health issues for mothers and their infants, including poor development and increased disease susceptibility [1]. The prevalence is higher in low-income countries, where access to diverse, nutritious foods is limited [6–8]. In regions like South Asia and Sub-Saharan Africa, lactating mothers face a heightened risk of nutritional deficiencies due to an increased physiological demand, the lactogenesis process, consuming monotonous diet, and increased dietary needs during breastfeeding [9].

Sub-Saharan African countries indeed bear a significant burden of global maternal morbidity and mortality, largely due to inadequate nutrition. The World Health Organization reports that in 2020, Sub-Saharan Africa alone accounted for approximately 70% of global maternal deaths, with most of these deaths being preventable and occurring in low and lower middle-income countries and these risks are exacerbate by nutritional inadequacies [10]. Dietary diversity among lactating mothers in Sub-Saharan African countries, particularly in Mozambique, Ghana, Kenya, Tanzania, and Burkina Faso, is essential for maternal and child health but is often inadequate. Studies indicate that a significant number of women in these regions do not meet the minimum dietary diversity requirements, which can lead to nutritional deficiencies [2,11,12].

Inadequate nutrition among lactating mothers can lead to insufficient milk quality, depriving infants of essential nutrients necessary for optimal growth and development [13,14]. This shortfall can manifest as stunting and other growth issues in children, which are not merely physical but also impede cognitive abilities. The repercussions extend

beyond health, influencing a child's educational performance and future learning potential [9,15]. Consequently, these early deficits can culminate in diminished productivity and economic contributions in adulthood, perpetuating a cycle of poverty and social disadvantage. Addressing maternal nutrition is therefore crucial, not only for the immediate well-being of mother and child but also for the long-term socio-economic stability of communities [16, 17].

Short-term supplementation, medium-term food fortification, and a long-term focus on dietary diversification are the three main strategies that have been reported to be effective in preventing and controlling hidden hunger [18]. Many scientific findings support the expanding global agreement that women, especially lactating mothers, should eat a wider variety of foods to enhance the nutritional diversity of their diets and that of their infants [19, 20]. More food categories must be included in the daily diet in order to meet the body's nutrient requirements because no one food group can contain all the nutrients needed for the body to function and perform in an optimal way [21–23].

The second Sustainable Development Goal (SDG) aims to end hunger and all forms of malnutrition by 2030 [22]. Long-term and permanent solutions to end global hunger may be built on ensuring a varied and sufficient diet [24]. One of the most important causes of micronutrient deficiencies and macronutrient shortages is an inadequate minimum dietary diversity among lactating mothers [6,13].

Despite Sub-Saharan African countries sharing the huge burden of global maternal morbidity and mortality related to inadequate nutrition, the previous few studies focused at the institutional level, and using a small sample size from primary data. However, this study analyzed factors at the individual and community levels that have a greater impact on minimum dietary diversity among lactating mothers.

This study allows for a more comprehensive understanding of the factors that contribute to dietary diversity, which is essential for developing targeted interventions to improve maternal nutrition and, consequently, reduce morbidity and mortality rates in the region. The findings from such a study could be instrumental in informing policy and public health strategies to enhance the nutritional status of lactating mothers across Sub-Saharan Africa. Therefore, the present study aimed to investigating the minimum dietary diversity and its determinants among lactating mothers in five Sub-Saharan African countries using a multilevel mixed effect analysis of the most recent Demographic and Health Surveys from 2021–2023.

## Methods and materials

### Study setting

The sub-Saharan region of Africa, defined as the area south of the Sahara Desert, is divided into four vast and distinct regions: Eastern, Central, Western, and Southern Africa, covering 9.4 million square miles. Demographically, the region is experiencing rapid growth, with projections estimating around 407 million women of reproductive age by 2030, and this number is expected to increase to 607 million by 2050 [25, 26].

Burkina Faso is a landlocked country in West Africa. As of mid-2024, the population is estimated to be approximately 23.8 million [27]. In Burkina Faso, as of the latest reports from 2023, there were 1,034,163 pregnant and lactating women, including 713,709 in emergency areas. This number represented 114% of the 2023 target set for reaching out to this demographic with aid and support [28]. Ghana, located in West Africa and bordering the Gulf of Guinea and the Atlantic Ocean, has a population of approximately 34.8 million as of 2024 [29]. Nutritional deficiencies among lactating women in Ghana are indeed a concern, with issues such as low dietary diversity and inadequate micronutrient intake being prevalent [30].

Tanzania, located in East Africa within the African Great Lakes region. As of mid-2024, the population is estimated to be approximately 69.2 million [31]. The country faces the double burden of malnutrition among lactating women, with issues of both under-nutrition and over-nutrition affecting their children [32]. Kenya is a country in East Africa that borders the Indian Ocean and Lake Victoria. As of 2024, the population of Kenya is estimated to be approximately 56.2 million [33]. In Kenya, lactating women grapple with micronutrient deficiencies, particularly in iron, foliate, and vitamin A, impacting both maternal and child health [34].

Mozambique, located in East African country along the Indian Ocean. As of 2024, the population of Mozambique is estimated to be approximately 34.9 million [35]. Lactating women in the country face numerous challenges, including high rates of child stunting and widespread micronutrient deficiencies. These issues are compounded by barriers such as inadequate breastfeeding support and education, which are critical for both mother and child health [36].

The five Sub-Saharan African countries included in this study were selected based on the availability of critical data on maternal nutrition in their DHS datasets, specifically focusing on the outcome variable, minimum dietary diversity. These countries provided comprehensive maternal nutrition indicators essential for accurately assessing dietary diversity, such as detailed food group consumption data. However, many other DHS datasets lacked these essential variables, making them unsuitable for inclusion in the study.

## Study design and period

The Demographic and Health Surveys (DHS) program utilizes a multistage sampling design to collect data, which is ideal for multilevel analysis. It includes diverse datasets such as individual and household survey data, HIV test results, and geographic data. These datasets encompass a broad spectrum of information on health, nutrition, population demographics, and disease prevalence, which are crucial for policy-making and research in public health. For the purpose of this multilevel analysis, we focused on the DHS surveys of 2021, 2022, and 2023. The DHS employs a two-stage stratified sampling technique, which is ideal for multilevel modeling as it allows for the examination of both individual and community-level factors.

## Population and eligibility criteria

The study's population and eligibility criteria were specifically defined to include lactating mothers aged 15-49 residing in five sub-Saharan African countries: Burkina Faso, Ghana, Kenya, Mozambique, and Tanzania. The source population comprised all lactating mothers within this age range across these countries. For the purpose of the study, the study population was narrowed down to those lactating mothers who resided within the enumeration areas that were selected for inclusion in the analysis. This selection process was designed to ensure a representative sample of the lactating mother demographic within the specified age bracket, from varied geographical and socio-economic backgrounds in the targeted countries, providing a comprehensive overview for the research objectives.

## Data source and sampling procedure

In this study, the data source consisted of Demographic and Health Surveys (DHS) conducted in five sub-Saharan African countries: Burkina Faso, Ghana, Kenya, Mozambique, and Tanzania (S1). These surveys collect comprehensive information on key health indicators, including fertility, mortality, nutrition, maternal and child health, HIV/AIDS, and gender-based

violence. The sampling procedure involved a stratified two-stage cluster design. First, the DHS program identified enumeration areas (clusters) within each country.

Then, a sample of households was selected from each enumeration area. The study extracted dependent and independent variables from the individual record dataset (IR file) and combined data across countries using STATA/SE. The outcome variable, minimum dietary diversity, was recoded as inadequate and adequate based on the number of diverse food groups consumed by lactating women in the 24 hours before assessment, as determined by the minimum dietary diversity score (MDD-W). The women's dietary diversity score (MDD-W) is a key indicator based on a woman's consumption of 10 distinct food categories within a 24-hour recall period. Total weighted samples of 19,917 lactating mothers were included in the study (**Table 1**).

## Study variables

**Dependent variables.** The individual record (IR) data set was used to generate the study's outcome variable, minimum dietary diversity. The FAO's the minimum dietary diversity for a woman (MDD-W), a dichotomous indicator/tool, was used to measure the outcome variable (minimum dietary diversity). In this study, a total of ten food groups were taken into account: cereals, white roots, tubers, and plantains; dark green leafy vegetables; other vitamin A-rich vegetables and fruits; other vegetables; other fruits; animal-source foods (meat and poultry, fish and seafood, eggs, milk, and dairy products); legumes, nuts, and seeds; oils and fats; sweets; and beverages, condiments, and miscellaneous [22]. In order to categorize lactating mothers' dietary diversity as adequate (≥5 food groups) and inadequate (<5 food groups) from 10 food groups, the Minimum Dietary Diversity Score (MDDS) was computed for each mother during the preceding 24 hours. Finally, the outcome variable was recoded as: a minimum dietary diversity score ≥ 5 food categories as "1" and a minimum dietary diversity score < 5 food groups as "0" [37].

**Independent variables.** Since DHS data are hierarchical in nature; independent variables from two sources (variables at the individual and community levels) were used for this analysis.

**Level 1 or individual-level independent variables.** Maternal age (15-19, 20-34, 35-49), maternal educational status (no formal education, primary, secondary and higher), husband educational status (no formal education, primary, secondary and higher), maternal employment (not employed, employed), marital status of the mother (unmarried, married, widowed/divorced/separated), number of ANC visits (no visit, 1-3, ≥ 4), total children ever born (≤3, > 3), household wealth index (poor, middle, rich), distance to health facility (big problem, not big problem), household media exposure (no, yes), place of delivery (home, health institution), preceding birth interval (≤24 months, > 24 months).

**Level 2 or community-level independent variables.** Place of residence (urban or rural), community-level women's illiteracy (low, high), community-level poverty (low, high),

**Table 1. Sample size for minimum dietary diversity and its determinants among lactating mothers in five Sub-Saharan African countries, DHS 2021-2023.**

| Countries | Year of survey | Unweighted sample (n) | Weighted sample (n) | Weighted sample (%) |
|---|---|---|---|---|
| Burkina Faso | 2021 | 4377 | 4332 | 21.75 |
| Ghana | 2022 | 3396 | 2925 | 14.69 |
| Kenya | 2022 | 6697 | 5974 | 29.99 |
| Mozambique | 2022/23 | 3037 | 3199 | 16.06 |
| Tanzania | 2022 | 3463 | 3487 | 17.51 |
| Total sample size | | 20,970 | 19,917 | 100 |

community-level media exposure (low, high), and community-level ANC utilization (low, high), community-level institutional delivery (low, high) country (Burkina Faso, Ghana, Kenya, Tanzania, Mozambique) (**Fig 1**).

## Operational definition of variables

### Lactating mothers

Women who are currently breastfeeding or expressing milk to feed their infant, typically within the first two years postpartum, as recommended by the World Health Organization (WHO) for optimal maternal and child health outcomes [38, 39].

### Wealth index

The Wealth Index in the Demographic and Health Surveys (DHS) serves as a valuable measure for assessing a household's relative economic status. Constructed from existing survey

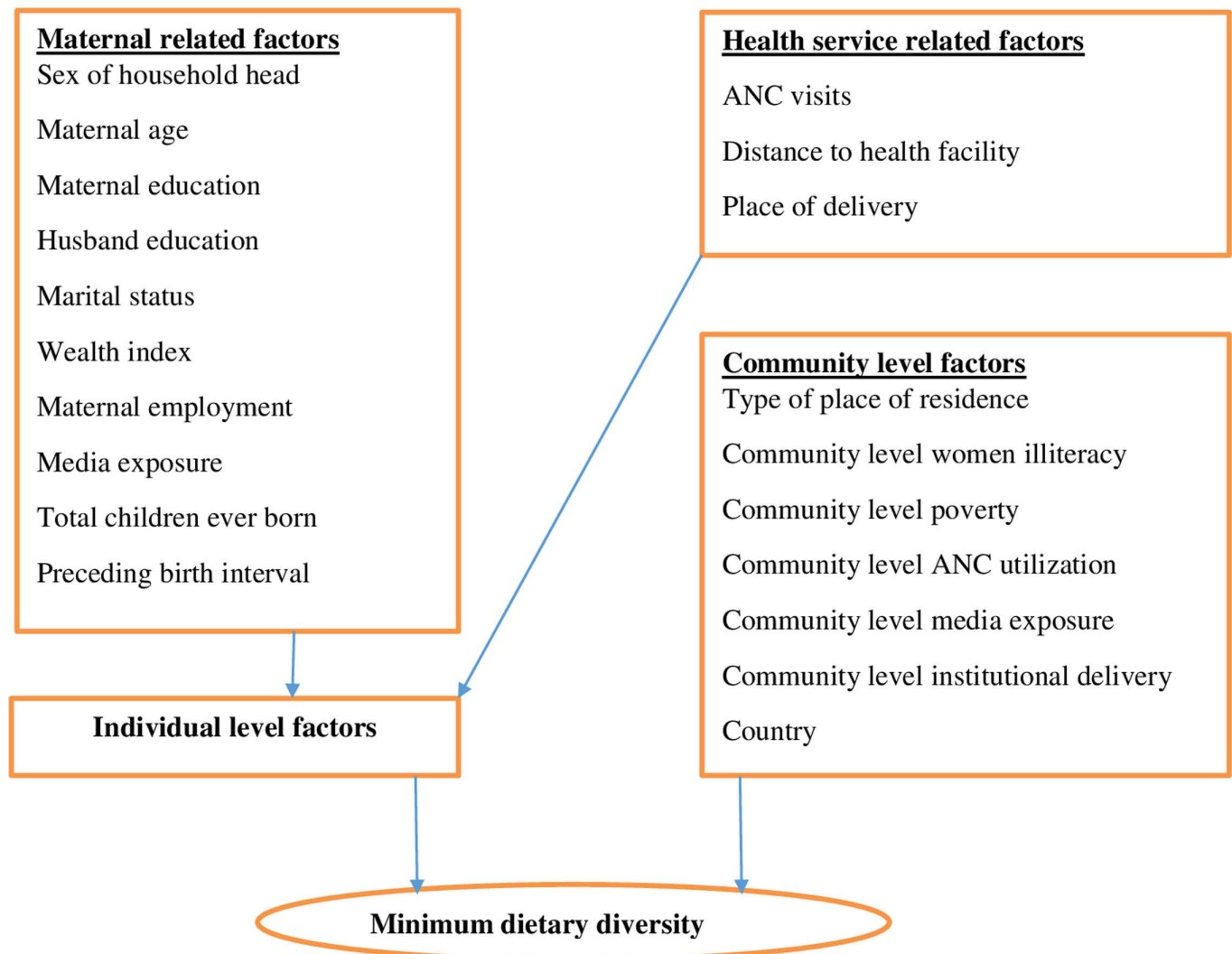

**Fig 1. Conceptual framework for factors associated with minimum dietary diversity among lactating mothers in five Sub-Saharan African countries.**

data, this index takes into account various factors, including household assets, services, and amenities. By analyzing these components, it provides insights into societal indicators such as health, nutrition, education, and population, all within the context of economic well-being. Essentially, the Wealth Index allows us to understand how households fare economically in relation to one another, aiding policymakers and researchers in making informed decisions [40].

### Community-level women illiteracy

Using data regarding respondents' educational attainment, the percentage of women with at least a primary education is calculated. Using the cluster number (v001), the individual level of women's education was cross-tabulated, and the national mean value was used for classification. Women's illiteracy at the community level was categorized as low in communities with ≥ 50% of the national mean value of women's education and high in communities with < 50% of the national mean value of women's community illiteracy [41].

### Community-level poverty

When producing it, the proportion of women in the rich and middle-class categories is taken into account. It was then classified using the national mean value of the wealth index as low community-level poverty (communities with ≥ 50% of the national mean value of the community wealth index) and high community-level poverty (communities with < 50% of the national mean value of the community wealth index) after the cross-tabulating individual-level combined wealth index with the cluster number (v001) was calculated [42].

### Data processing and statistical analysis

Following the extraction of data from current DHS data sets, STATA/SE version 17 statistical software was used to clean, input, and analyze the data. Before conducting any statistical analysis, the data were weighted using the sampling weight, primary sampling unit, and stratum in order to restore the representativeness of the survey and account for the sample design when computing standard errors to produce accurate statistical estimations. We used the weighting variable (v005) as a relative weight normalized to make the analysis survey-specific.

The weight adjusted for women = V005 × (total number of lactating mothers in the entire country between the ages of 15 and 59 at the time of the survey)/ (number of lactating mothers aged 15-49 years in the survey) is how we denormalized the individual standard weight of lactating mothers for the pooled data by dividing it by the sample proportion of each country. The standard logistic regression model's assumptions of equal variance and independence observations are broken by the hierarchical nature of the DHS data. The independent observations and equal variance assumptions between clusters of the ordinal logistic regression model are broken when lactating mothers are nested within a cluster and we presume that study subjects in that cluster might share characteristics with participants in another cluster. This implies the need for a sophisticated model that accounts for between-cluster factors.

In light of this, multilevel mixed-effects logistic regression has been used to identify factors associated with minimum dietary diversity among lactating mothers. A binary logistic regression model was employed to identify the contributing factors to minimum dietary diversity. The factors contributing minimum dietary diversity have been reported as an adjusted odds ratio (AOR) with a 95% significance level. A p-value of less than 0.25 in the biivariable analysis indicated that the data would be a good fit for the multivariable analysis. In multivariable logistic regression, all variables with p values < 0.05 were deemed statistically significant.

## Model building for multilevel analysis

Multilevel analysis, also known as hierarchical or mixed-effects modeling, is a statistical method designed to handle the nested structure of data, where observations are grouped within higher-level units such as individuals within households, households within communities, or regions within countries. This method was selected due to the hierarchical nature of our data, where individual lactating mothers are nested within households, which in turn are nested within communities or regions. Multilevel modeling addresses the non-independence of observations within these clusters by partitioning variance at each level and accounting for contextual factors at higher levels, such as community or regional characteristics. It simultaneously analyzes individual-level factors, leading to more precise estimates and a deeper understanding of the determinants of breastfeeding practices. By modeling the data's hierarchical structure, multilevel analysis effectively separates individual effects from contextual influences, providing a comprehensive framework to capture dependencies and variability across different levels.

$$\text{Logit}\left(Y_{ij}\right) = \beta_0 + \beta_1 X_{ij} + u_{0j} + e_{ij} \,(41), \text{where}$$

$Y_{ij}$: Binary outcome for individual (i) in cluster (j).
$X_{ij}$: Individual-level predictors.
$u_{0j}$: Random intercept for cluster (j).
$e_{ij}$: Individual-level error term.

## Random effects

To assess the variation of minimum dietary diversity among lactating mothers between clusters, random effects or measures of variation, such as the Likelihood Ratio test (LR), Intra-class Correlation Coefficient (ICC), and Median Odds Ratio (MOR), were taken into account. The ICC measures the amount of variance in minimum dietary diversity that can be attributed to differences across clusters, or the degree of heterogeneity of minimum dietary diversity between clusters, using clusters as a random variable. This was calculated with the following formula: $\text{ICC} = \dfrac{VC}{VC + 3.29} \times 100\%$ [43].

The Median Odds Ratio (MOR) measures the variation in minimum dietary diversity between clusters in terms of odds ratio and is defined as the median value of the odds ratio between the cluster at high likelihood of minimum dietary diversity and the cluster at lower risk when individuals are randomly selected from two clusters: $\text{MOR} = e^{0.95\sqrt{VC}}$ [44].

Furthermore, the PCV, which is computed as $\text{PCV} = \text{PCV} = \dfrac{Vnull - Vc}{Vnull} \times 100\%$ [45],

demonstrates how differences in insufficient dietary diversity are explained by determinants. Here, Vnull denotes the variance of the null model and VC stands for cluster level variance.

Fixed effects were used to quantify the relationship between the likelihood of minimum dietary diversity and independent variables at the individual and community levels. The adjusted odds ratio (AOR) and 95% confidence intervals were utilized to assess it and demonstrate its strength with a p-value of less than 0.05. The models were compared using the log likelihood ratio and deviance = -2 (log likelihood ratio) due to the nested nature of the data; the model with the highest log likelihood ratio and the lowest deviance was chosen as the best-fit model. The multi-collinearity was evaluated by computing the variance inflation factors (VIF).

STROBE Statement—checklist (S2)

### Ethical approval and consent to participate

Since this study is purely a secondary review of the DHS data, ethical approval is not needed. We enrolled with the DHS web archive, requested the dataset for our study, and were granted permission to view and download the data files. As per the DHS study, all participant data were anonymized at the time of survey data collection. Visit http://www.dhsprogram.com for additional information on DHS data and ethical standards.

## Result

### Socio-demographic and economic characteristics of lactating mothers in five Sub-Saharan African countries, DHS 2021-2023

A total of 19,917 lactating mothers were included in this study. Nearly one-third of lactating mothers 5,983 (30.04%) had no formal education. Around one-fifth 3720 (18.68%) of lactating mothers delivered at home, and about 6,058 (30.42%) were living in rural areas of sub-Saharan Africa. About 1,122 (5.63%) of lactating mothers did not have ANC visits. More than half (53.36%) of lactating mothers had low access to community media exposure (**Table 2**).

### Minimum dietary diversity among lactating mothers in five Sub-Saharan African countries

The magnitude of minimum dietary diversity among lactating mothers in Burkina Faso, Ghana, Kenya, Mozambique, and Tanzania was 25.66% (95% CI: 24.47, 25.75). The magnitude of urban and rural minimum dietary diversity in five sub-Saharan African countries was found to be 14.01% and 11.04%, respectively (**Fig 2**). 18.48% of Burkina Faso, 53.44% of Ghana, 33.10% of Kenyans, 13.05% of Mozambique, and 10.21% of Tanzanian lactating mothers meet the criteria for minimum dietary diversity (**Fig 3**).

### Measures of variation and model fitness

The results from the null model, which exhibited a variance of 0.4487294, revealed significant disparities in dietary diversity across communities. Approximately 12.00% of the overall variation in minimum dietary diversity within the null model was attributable to community-level factors at the cluster level. Notably, the null model yielded the highest Median Odds Ratio (MOR) value of 1.89. This implies that individuals randomly selected from a high cluster for minimum dietary diversity are 1.89 times more likely to have such minimum dietary diversity compared to their counterparts in a inadequate minimum dietary diversity cluster.

Model I's intraclass correlation value indicated that 6.80% of the variation in minimum dietary diversity could be attributed to differences between communities. Subsequently, we developed Model II, incorporating community-level variables alongside the null model. Based on the Intraclass Correlation Coefficient (ICC) value from Model II, cluster variations accounted for 8.37% of the variance in minimum dietary diversity.

In the final model (Model III), both individual and community-level factors contributed to approximately 42.35% of the variation in the likelihood of minimum dietary diversity. Model III, characterized by the lowest deviance (11,455.13) and the highest log-likelihood ratio (-5727.565), emerged as the best-fitted model (**Table 3**).

### Factors associated with minimum dietary diversity among lactating mothers in five Sub-Saharan African countries

In a multivariable multilevel mixed-effect logistic regression analysis, where both individual and community-level factors were simultaneously considered, maternal educational level,

**Table 2. Socio-demographic and economic characteristics of lactating mothers in five Sub-Saharan African countries, DHS 2021-2023.**

| Variables | Frequency (n) | Percent (%) |
|---|---|---|
| **Individual level factors** | | |
| Sex of household head | | |
| Male | 15,569 | 78.17 |
| Female | 4,348 | 21.83 |
| Maternal age | | |
| 15-19 | 1982 | 9.95 |
| 20-34 | 14014 | 70.36 |
| 35-49 | 3921 | 19.69 |
| Maternal educational level | | |
| No formal Education | 5,983 | 30.04 |
| Primary Education | 6,735 | 33.82 |
| Secondary and higher | 7,199 | 36.14 |
| Husband educational level | | |
| No formal Education | 5,734 | 33.47 |
| Primary Education | 5,154 | 30.08 |
| Secondary and higher | 6,245 | 36.45 |
| Maternal employment status | | |
| Not employed | 8,801 | 44.19 |
| Employed | 11,116 | 55.81 |
| Marital status of the mother | | |
| Never married | 1,592 | 7.99 |
| Currently married | 17,181 | 86.26 |
| Divorced/Widowed/Separated | 1,144 | 5.75 |
| Distance to health facility | | |
| Big problem | 6,091 | 35.81 |
| Not a big problem | 10,920 | 64.19 |
| Place of delivery | | |
| Home | 3720 | 18.68 |
| Health institution | 16197 | 81.32 |
| Number of ANC visits | | |
| No visit | 1,122 | 5.63 |
| 1-3 | 5,350 | 26.86 |
| ≥4 | 13,445 | 67.50 |
| Total children ever born | | |
| ≤3 | 12,528 | 62.90 |
| >3 | 7,389 | 37.10 |
| Preceding birth interval | | |
| ≤24 months | 2,068 | 13.75 |
| >24 months | 12,972 | 86.25 |
| Household wealth index | | |
| Poor | 9,007 | 45.22 |
| Middle | 3,872 | 19.44 |
| Rich | 7,038 | 35.33 |
| Household media exposure | | |
| No | 6,231 | 31.28 |
| Yes | 13,686 | 68.72 |

*(Continued)*

**Table 2.** (Continued)

| Variables | Frequency (n) | Percent (%) |
|---|---|---|
| **Community level factors** | | |
| Place of residence | | |
| Rural | 6,058 | 30.42 |
| Urban | 13,859 | 69.58 |
| Community media exposure | | |
| Low | 10,628 | 53.36 |
| High | 9,289 | 46.64 |
| Community poverty | | |
| Low | 9,426 | 47.33 |
| High | 10,491 | 52.67 |
| Community women's' illiteracy | | |
| Low | 8,534 | 42.85 |
| High | 11,383 | 57.15 |
| Community ANC utilization | | |
| Low | 8,132 | 40.83 |
| High | 11,785 | 59.17 |
| Community institutional delivery | | |
| Low | 9,193 | 46.16 |
| High | 10,724 | 53.84 |
| Country | | |
| Burkina Faso | 4,332 | 21.75 |
| Ghana | 2,925 | 14.69 |
| Kenya | 5,974 | 29.99 |
| Mozambique | 3,199 | 16.06 |
| Tanzania | 3,487 | 17.51 |

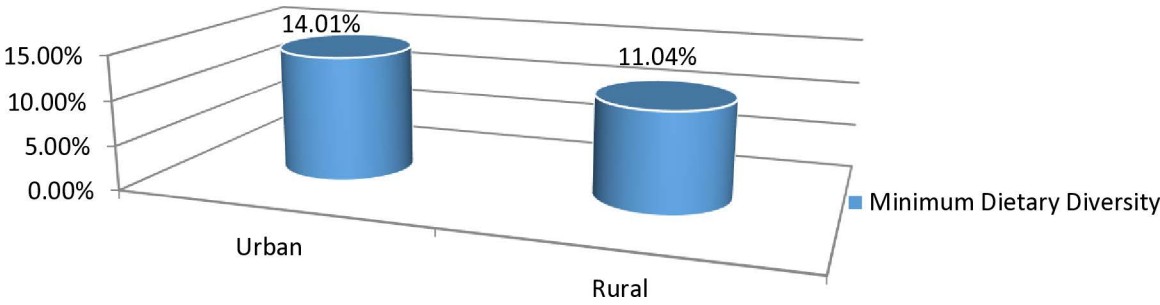

**Fig 2. Urban and rural magnitude of minimum Dietary Diversity among lactating mothers in five Sub-Saharan African countries, DHS 2021-2023.**

maternal employment status, distance to health facilities, health facility delivery, rich wealth status, community ANC utilization, and reside in Ghana were found to be significantly associated with achieving minimum dietary diversity among lactating mothers.

Lactating mothers whose household head was male had 1.15 times higher odds of achieving minimum dietary diversity compared to their counterparts (AOR = 1.15, 95% CI: 1.00, 1.31). The odds of dietary diversity was 1.38 times higher among lactating mothers who attained secondary

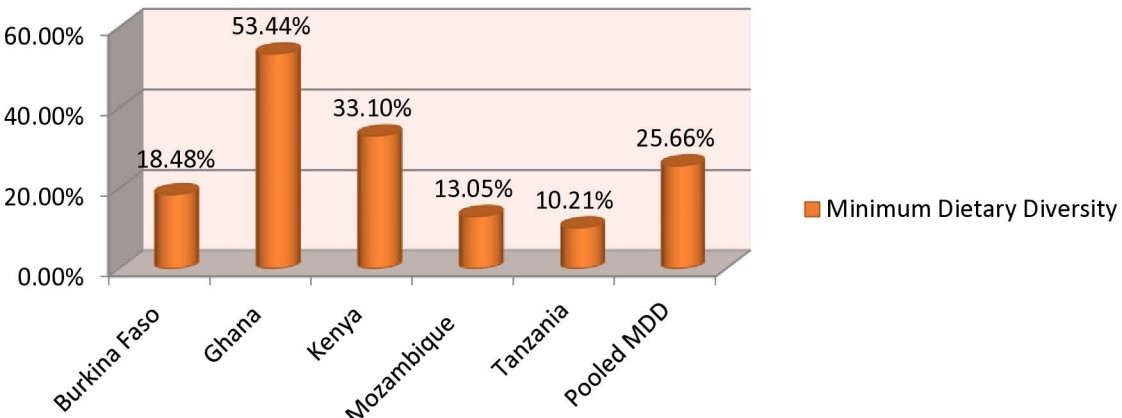

**Fig 3. Minimum dietary diversity among lactating mothers in five Sub-Saharan African countries, DHS 2021-2023.**

Table 3. Model comparison and random effect analysis for minimum Dietary Diversity among lactating mothers in five Sub-Saharan African countries, DHS 2021-2023.

| Parameter | Null model | Model I | Model II | Model III |
|---|---|---|---|---|
| Variance | 0.4487294 | 0.2401967 | 0.3003593 | 0.2587025 |
| ICC | 12.00% | 6.80% | 8.37% | 7.29% |
| MOR | 1.89 | 1.59 | 1.68 | 1.62 |
| PCV | Reference | 46.47% | 33.06%. | 42.35%. |
| **Model fitness** | | | | |
| LLR | -11607.226 | -6108.0594 | -10500.575 | -5727.565 |
| Deviance | 23.214.088 | 12,214.1188 | 21,001.15 | 11,455.13 |

ICC: interacluster correlation, LLR: logliklihood ratio, MOR: median odds ratio, PCV: proportional change in variance.

education or higher compared with lactating mothers who had no formal education (AOR = 1.38, 95% CI: 1.18, 1.61). Lactating mothers whose husbands have attained secondary education or higher had 1.38 times greater odds of achieving minimum dietary diversity compared to women whose husbands had no formal education. Employed lactating mothers exhibited 1.40 times greater odds of achieving minimum dietary diversity compared to their unemployed counterparts (AOR = 1.40, 95% CI: 1.26, 1.56). Lactating mothers with no significant distance barrier to health facilities had 1.35 times greater odds of achieving minimum dietary diversity compared to mothers facing substantial distance challenges (AOR = 1.35, 95% CI: 1.21, 1.51).

The odds of minimum dietary diversity were 1.25 times higher among lactating mothers who gave birth at health facilities compared to mothers who delivered at home (AOR = 1.25, 95% CI: 1.08, 1.45). Lactating mothers from higher wealth quantiles exhibited 1.86 times greater odds of achieving minimum dietary diversity compared to mothers from poorer wealth status (AOR = 1.86, 95% CI: 1.60, 2.17). Lactating mothers living in less impoverished communities had 1.16 times greater odds of achieving dietary diversity compared to those in more impoverished communities (AOR = 1.16, 95% CI: 1.01, 1.34). Lactating mothers with high community-level ANC utilization had 1.18 times greater odds of achieving minimum dietary diversity compared to mothers with low community-level ANC utilization (AOR = 1.18, 95% CI: 1.04, 1.35). The odds of minimum dietary diversity were 4.21 times higher among lactating mothers who reside in Ghana compared to lactating mothers from Burkina Faso (AOR = 4.21, 95% CI: 3.60, 4.94) (**Table 4**).

**Table 4. Multivariable multilevel logistic regression analysis of individual-level and community level determinants of minimum dietary diversity among lactating mothers in five Sub-Saharan African countries, DHS 2021-2023.**

| Individual level variables | Model I AOR(95% CI) | Model II AOR(95% CI) | Model III AOR(95% CI) |
|---|---|---|---|
| Sex of household head | | | |
| Male | 1.03(0.91, 1.17) | | 1.15(0.89, 1.31) |
| Female | 1 | | 1 |
| Maternal age | | | |
| 15-19 | 0.63(0.42, 0.96) | | 0.67(0.44, 1.02) |
| 20-34 | 1 | | 1 |
| 35-49 | 1.21(1.08, 1.36) | | 1.12(0.99, 1.26) |
| Maternal educational level | | | |
| No formal Education | 1 | | 1 |
| Primary Education | 0.88(0.77, 1.00) | | 0.98(0.85, 1.13) |
| Secondary and above | 1.66(1.43, 1.92) | | **1.38(1.18, 1.61)** |
| Husband educational level | | | |
| No formal Education | 1 | | 1 |
| Primary Education | 0.77(0.68, 0.89) | | 0.98(0.84, 1.13) |
| Secondary and above | 1.52(1.32, 1.75) | | 1.21(0.64, 1.40) |
| Maternal employment | | | |
| Not employed | 1 | | 1 |
| Employed | 1.71(1.54, 1.89) | | **1.40(1.26, 1.56)** |
| Distance to health facility | | | |
| Big problem | 1 | | 1 |
| Not a big problem | 1.41(1.27, 1.57) | | **1.35(1.21, 1.51)** |
| Place of delivery | | | |
| Home | 1 | | 1 |
| Health institution | 1.23(1.07, 1.41) | | **1.25(1.08, 1.45)** |
| Number of ANC visits | | | |
| No visit | 1 | | 1 |
| 1-3 | 1.03(0.80, 1.32) | | 0.91(0.70, 1.18) |
| ≥4 | 1.26(0.99, 1.61) | | 0.92(0.72, 1.19) |
| Total children ever born | | | |
| ≤3 | 1.04(0.94, 1.16) | | 1.06(0.95, 1.18) |
| >3 | 1 | | 1 |
| Household wealth index | | | |
| Poor | 1 | | 1 |
| Middle | 0.98(0.86, 1.11) | | 1.33(0.16, 1.53) |
| Rich | 1.03(0.91, 1.16) | | **1.86(1.60, 2.17)** |
| **Preceding birth interval** | | | |
| ≤24 months | 1 | | 1 |
| >24 months | 1.04(0.90, 1.19) | | 1.03(0.89, 1.19) |
| Household media exposure | | | |
| No | 1 | | 1 |
| Yes | 2.02(1.79, 2.27) | | 1.75(1.54, 1.98) |
| **Community level variables** | | | |
| Place of residence | | | |
| Rural | | 1 | 1 |

*(Continued)*

**Table 4.** (Continued)

| Individual level variables | Model I AOR(95% CI) | Model II AOR(95% CI) | Model III AOR(95% CI) |
|---|---|---|---|
| Urban | | 1.71(1.57, 1.86) | 0.94(0.82, 1.07) |
| Community media exposure | | | |
| Low | | 1 | 1 |
| High | | 1.27(1.12, 1.45) | 1.07(0.92, 1.25) |
| Community poverty | | | |
| Low | | 1.40(1.25, 1.56) | 1.16(0.81, 1.34) |
| High | | 1 | 1 |
| Community women's illiteracy | | | |
| Low | | 0.75(0.66, 0.85) | 0.90(0.77, 1.05) |
| High | | 1 | 1 |
| Community ANC utilization | | | |
| Low | | 1 | 1 |
| High | | 1.09(0.98, 1.22) | **1.18(1.04, 1.35)** |
| Community institutional delivery | | | |
| Low | | 1 | 1 |
| High | | 1.18(1.06, 1.31) | 1.07(0.93, 1.22) |
| Country | | | |
| Burkina Faso | | 1 | 1 |
| Ghana | | 4.39(3.93, 4.89) | **4.21(3.60, 4.94)** |
| Kenya | | 1.35(1.21, 1.51) | 1.38(0.16, 1.65) |
| Mozambique | | 0.75(0.66, 0.85) | 0.96(0.80, 1.15) |
| Tanzania | | 0.46(0.40, 1.52) | 0.43(0.36, 1.53) |

## Discussion

Minimum Dietary Diversity (MDD) during breastfeeding plays a vital role in maternal and child health. When lactating mothers maintain a diverse diet, it directly enhances the nutritional content of breast milk. This rich milk provides essential nutrients such as vitamins, minerals, proteins, and healthy fats to support optimal growth and development in infants. Ensuring MDD benefits both mother and child, fostering overall well-being [46]. According to the 2016 FAO guideline, in order to have optimal micronutrient sufficiency, all women should meet the minimum dietary diversity score of ≥ 5 food groups [47].

This study aimed to assess the prevalence and determinants of minimum dietary diversity among lactating mothers in Burkina Faso, Ghana, Kenya, Mozambique, and Tanzania using 2021, 2022 and 2023 Demography and Health Survey data from each country. In this study, the magnitude of minimum dietary diversity among lactating mothers in five Sub-Saharan countries was 25.66% (95% CI: 24.47, 25.75). This finding is consistent with the study conducted in South Africa (25%) [48]. It is lower than the study findings conducted in Ethiopia (48.8%) [37], India (77.1%) [1], and Bangladesh (29.7% [3]. The lower prevalence of minimum dietary diversity among lactating mothers in Burkina Faso, Ghana, Kenya, Mozambique, and Tanzania compared to Ethiopia, India, and Bangladesh. Differences in dietary diversity across countries can indeed be influenced by various cultural, economic, and health system factors. For instance, in Sub-Saharan Africa, traditional dietary patterns, limited access to diverse food sources, and lower economic resources contribute to lower dietary diversity among lactating mothers compared to South Asia, where a wider availability of food, higher

maternal education, and greater awareness of nutrition may promote better dietary practices [49]. Cultural factors also play a significant role, as food taboos and preferences in certain countries may limit the variety of foods consumed during lactation[50]. Additionally, variations in maternal meal frequency, antenatal care, and paternal education could contribute to the disparities observed across these countries [23,51]. In contrast the prevalence of minimum dietary diversity in this study is higher than the study conducted in Vietnam [52]. The inconsistency may be attributed to differences in sample size, as the Vietnam study relied on primary data with a small sample size, while the current study utilized large surveys from DHS.

Maternal education level, maternal employment status, distance to health facilities, health facility delivery, rich wealth status, community ANC utilization, and residence in Ghana were found to be significantly associated with achieving minimum dietary diversity among lactating mothers in a multivariable multilevel mixed-effect logistic regression analysis.

The odds of dietary diversity was 1.38 times higher among lactating mothers who attained secondary education or higher compared with lactating mothers who had no formal education. It is in line with study findings in Ethiopia [5,37,53] and India [1]. The higher odds of dietary diversity among lactating mothers with secondary education or higher can be attributed to several factors. Firstly, education often correlates with better awareness of nutrition and dietary requirements, leading to more informed food choices [2]. Secondly, educated mothers may have greater access to diverse food sources due to improved socioeconomic status and knowledge about balanced diets [54]. Lastly, education can positively influence health-seeking behavior, including seeking advice on nutrition and dietary practices, which contributes to better dietary diversity [1]. Lactating mothers with no significant distance barrier to health facilities had 1.35 times greater odds of achieving minimum dietary diversity compared to mothers facing substantial distance challenges. It is supported by the studies conducted in Ethiopia [55, 56] and Switzerland [13]. The possible explanation could be when health facilities are easily accessible; women are more likely to receive education on nutrition during pregnancy and lactation. This vital information includes the importance of a diverse diet and micronutrient-rich foods. Additionally, shorter distances facilitate more frequent postnatal care visits, allowing healthcare providers to assess and provide advice on nutritional status effectively. Conversely, women facing substantial distance challenges may miss out on these opportunities, leading to lower dietary diversity [37,57].

The odds of minimum dietary diversity were 1.25 times higher among lactating mothers who gave birth at health facilities compared to mothers who delivered at home. It is consistent with studies conducted in Malawi [58] and Italy [59]. The higher minimum dietary diversity among lactating mothers who delivered at health facilities compared to those who delivered at home can be attributed to several factors. Firstly, health facilities provide nutritional counseling and education during postpartum stays, emphasizing the importance of diverse diets. Secondly, access to balanced meals in health facilities ensures that mothers receive adequate nutrients during lactation. Thirdly, mothers who give birth at health facilities tend to have better awareness of dietary needs and are more likely to make informed choices. Lastly, the supportive environment in health facilities allows for discussions with healthcare providers, further influencing dietary habits [60, 61].

Lactating mothers from higher wealth quantiles exhibited 1.86 times greater odds of achieving minimum dietary diversity compared to mothers from poorer wealth status. This is supported by the previous studies conducted Ataye, Ethiopia [37], South Africa [48], and Ghana [62]. This phenomenon can be attributed to several factors: Access to Diverse Foods: Wealthier households often have better access to a variety of foods. They can afford a wider range of fruits, vegetables, proteins, and other nutrient-rich options, which contributes to dietary diversity [63]. Education and Awareness: Higher socioeconomic status is associated

with better education and awareness about nutrition. Wealthier mothers may have more knowledge about the importance of diverse diets during lactation and are more likely to make informed food choices [47]. Cultural Practices: Wealthier families may follow cultural practices that prioritize diverse diets. They might have access to traditional recipes and cooking methods that incorporate a wide range of food groups [64]. Food Security: Economic stability reduces food insecurity. When households are financially secure, they can consistently provide a variety of foods, ensuring better dietary diversity [65].

Lactating mothers with high community-level ANC utilization had 1.18 times greater odds of achieving minimum dietary diversity compared to mothers with low community-level ANC utilization. This is consistent with the studies conducted in Tanzania [66], India [67], and The WHO 2016 maternal nutrition programming [68]. The association between high community-level antenatal care (ANC) utilization and improved dietary diversity among lactating mothers can be attributed to several factors. ANC visits provide essential nutrition education and counseling, emphasizing the importance of a balanced diet during pregnancy and lactation. These sessions inform women about nutrient-rich foods, including vegetables, meat, dairy products, and fruits, which contribute to dietary diversity [68].

The odds of minimum dietary diversity were 4.21 times higher among lactating mothers who reside in Ghana compared to lactating mothers from Burkina Faso. The significantly higher odds of achieving minimum dietary diversity among lactating mothers in Ghana compared to those in Burkina Faso can be attributed to various factors. These include differences in cultural practices, food availability, and healthcare infrastructure. Ghana has made substantial progress in maternal and child health, with improved access to ANC services, nutrition education, and community-based interventions. Additionally, Ghana's diverse food culture, including staples like plantains, cassava, and fish, contributes to better dietary diversity. In contrast, Burkina Faso faces challenges related to food insecurity, limited healthcare access, and cultural norms that may impact dietary choices [69, 70].

The study's strength was its utilization of current large-sample national demography and health surveys from five sub-Saharan African nations. Another strength of this study was the use of mixed-effect multilevel logistic regression to identify two-level factors (individual and community-level factors), which was not possible with ordinary logistic regression. However, the study was unable to include these variables that might have been associated with the outcome variable because a number of important variables, such as food security and maternal attitude, were absent from the DHS data collection. Additionally, the chance to include DHS data from other countries in this research is limited by the lack of maternal nutrition-related data in many sub-Saharan African DHS data sets. Furthermore, data from different countries in the DHS may have been collected during varying time periods, which could introduce seasonality effects that may influence food availability and consumption patterns. Moreover, he potential for recall bias in dietary data, as mothers may forget or misreport details even within the 24-hour recall period. Additionally, while the Minimum Dietary Diversity (MDD) indicator is useful for population-level analysis, it may not accurately reflect an individual's typical dietary habits, as dietary patterns can fluctuate daily and may not represent usual intake at the household level.

## Conclusion and recommendations

The minimum dietary diversity among lactating mothers is low, according to this study's findings. The study found that the minimum dietary diversity was influenced by both community-level and individual-level variables. We recommend that the health ministries of Burkina Faso, Kenya, Ghana, Mozambique, and Tanzania implement targeted policies such as mobile health services for rural women, community-based health education programs, and financial incentives for pregnant women to attend antenatal visits. In countries like Sub-Saharan Africa,

existing educational policies could be integrated with maternal health programs to ensure that women, particularly those with low education levels, receive information on the importance of antenatal care. Additionally, partnerships with local organizations and community leaders can help reach marginalized populations and increase service utilization.

## Supporting information

**S1. Data. Data used for analysis.**
(RAR)

**S2 Data. Strobe-checklist.**
(DOCX)

## Author contributions

**Conceptualization:** Alebachew Ferede Zegeye, Desale Bihonegn Asmamaw, Desalegn Anmut Bitew, Lemlem Daniel Baffa, Melak Jejaw.

**Data curation:** Alebachew Ferede Zegeye, Tadesse Tarik Tamir, Bewuketu Terefe, Kaleb Assegid Demissie.

**Formal analysis:** Alebachew Ferede Zegeye, Tadesse Tarik Tamir, Elsa Awoke Fentie, Rahel Mulatie Anteneh, Misganaw Guadie Tiruneh, Tadele Biresaw Belachew.

**Investigation:** Alebachew Ferede Zegeye, Desalegn Anmut Bitew, Tadele Biresaw Belachew, Melak Jejaw.

**Methodology:** Alebachew Ferede Zegeye, Tadesse Tarik Tamir, Desale Bihonegn Asmamaw, Elsa Awoke Fentie, Bewuketu Terefe, Lemlem Daniel Baffa, Wubshet D. Negash.

**Project administration:** Tadele Biresaw Belachew.

**Resources:** Desalegn Anmut Bitew, Kaleb Assegid Demissie, Melak Jejaw.

**Software:** Alebachew Ferede Zegeye, Desale Bihonegn Asmamaw, Desalegn Anmut Bitew, Rahel Mulatie Anteneh, Misganaw Guadie Tiruneh, Wubshet D. Negash.

**Supervision:** Kaleb Assegid Demissie.

**Validation:** Alebachew Ferede Zegeye, Misganaw Guadie Tiruneh, Tadele Biresaw Belachew.

**Visualization:** Elsa Awoke Fentie, Misganaw Guadie Tiruneh.

**Writing – original draft:** Alebachew Ferede Zegeye, Tadesse Tarik Tamir, Desale Bihonegn Asmamaw, Desalegn Anmut Bitew, Elsa Awoke Fentie, Bewuketu Terefe, Rahel Mulatie Anteneh, Lemlem Daniel Baffa, Misganaw Guadie Tiruneh, Kaleb Assegid Demissie, Tadele Biresaw Belachew, Wubshet D. Negash, Melak Jejaw.

**Writing – review & editing:** Alebachew Ferede Zegeye, Tadesse Tarik Tamir, Desale Bihonegn Asmamaw, Desalegn Anmut Bitew, Bewuketu Terefe, Lemlem Daniel Baffa, Wubshet D. Negash.

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
