## [Decision Letter · Decision Letter 0]

20 Nov 2024

PONE-D-24-31238Minimum Dietary Diversity and its Determinants among Lactating Mothers in Five Sub-Saharan African countries: A multilevel analysisPLOS ONE

Dear Dr. Alebachew Ferede Zegeye,

Thank you for submitting your manuscript to PLOS ONE. After careful consideration, we feel that it has merit but does not fully meet PLOS ONE’s publication criteria as it currently stands. Therefore, we invite you to submit a revised version of the manuscript that addresses the points raised during the review process. A recurring comment has been the use of 16 food groups while 10groups are recommended. Authors should please give a justification and cite relevant literature. Authors should kindly take note and correct some errors and typos throughout the manuscript 

We look forward to receiving your revised manuscript.

Kind regards,

Engelbert A. Nonterah, MD, PhD

Academic Editor

PLOS ONE

Journal Requirements:

Reviewers' comments:

Reviewer's Responses to Questions

**Comments to the Author**

1. Is the manuscript technically sound, and do the data support the conclusions?

Reviewer #1: Partly

Reviewer #2: Yes

2. Has the statistical analysis been performed appropriately and rigorously? 

Reviewer #1: Yes

Reviewer #2: Yes

3. Have the authors made all data underlying the findings in their manuscript fully available?

Reviewer #1: Yes

Reviewer #2: Yes

4. Is the manuscript presented in an intelligible fashion and written in standard English?

Reviewer #1: Yes

Reviewer #2: Yes

5. Review Comments to the Author

Reviewer #1: This manuscript addresses an important issue regarding dietary diversity among lactating mothers in Sub-Saharan Africa, employing a strong analytical approach. However, improvements in justification, clarity, and actionable recommendations are necessary. Expanding on the policy implications and addressing regional disparities more effectively could enhance the study's impact.

Specific Major Comments

Abstract

The abstract provides a concise summary of the study, covering the background, methods, results, and conclusions. However, the following improvements could be made:

• Include the actual coverage of Minimum Dietary Diversity (MDD) with confidence intervals.

• While key findings are reported, the specific categories of the association between certain factors and dietary diversity could be more clearly articulated. For example, it is unclear how the distance to a health facility affects dietary diversity (DD). Does it refer to a long or short distance, or a significant or minor problem? In addition, when comparing secondary or higher education with no formal education, the reference group is incorrectly identified, as there is another level: primary education. Clarify if the place of delivery refers to home or another location.

• The results, such as the proportion of lactating mothers achieving dietary diversity, could benefit from contextualisation. Comparing dietary diversity across countries within the abstract could better summarise national differences.

Introduction

The introduction thoroughly addresses the importance of dietary diversity among lactating mothers and situates the study within a broader public health context. However, it could be improved by:

• Providing a clearer rationale for selecting the five Sub-Saharan African countries. A stronger justification for their inclusion based on specific nutritional challenges is needed. While the authors mention that women in these countries do not meet the minimum dietary diversity, this does not sufficiently justify the selection of these countries over others in Sub-Saharan Africa that face similar issues.

• There is limited discussion of gaps in the existing literature on multilevel determinants. Highlighting where this study contributes new insights would better position its relevance. Pooling data alone does not offer an opportunity to address country-specific determinants, particularly since the included countries are not representative of the entire Sub-Saharan African region or any specific sub-region of the continent.

Methods

The methodology is robust, utilising multilevel mixed-effects logistic regression to account for both individual and community-level factors. However, there are a few points of concern:

• The study uses DHS data from five countries, but the selection process for these countries is not fully justified. Providing a clearer explanation of why these countries were chosen over others would strengthen the credibility of the findings. Simply using the most recent datasets is insufficient justification.

• The term "lactating mothers" requires a clearer definition within the study.

• The FAO's MDD-W (latest version) includes ten distinct food groups. It is unclear how the authors arrived at 16 food groups for their analysis. Additionally, the acceptable dietary diversity cut-off is five out of ten food groups. Using the same cut-off for 16 food groups requires justification, as this results in approximately 30% coverage rather than the intended 50%.

• The conceptual framework does not adequately show how background factors like education or household socioeconomic status influence healthcare factors such as antenatal care utilisation or place of delivery.

• While the multilevel modelling approach is appropriate, the justification for selecting this method could be better explained for non-expert readers.

Discussion, conclusions, and limitations

• While some comparisons are made with studies from Ethiopia, India, and Bangladesh, these are largely limited to prevalence statistics. The authors should elaborate on why dietary diversity levels might differ in these countries, considering cultural, economic, and health system differences.

• The DHS data for the various countries may not have been collected in the same period. Seasonality, which affects food availability and consumption patterns, is a known factor. Comparing country-level dietary diversity without accounting for this limitation weakens the analysis.

• The policy implications of the findings could be expanded to provide specific recommendations for the five countries studied. Although the study mentions that ministries should focus on antenatal care utilisation and education, it lacks concrete policy suggestions or frameworks. For example, Ghana has existing policies to promote education, which could be better addressed.

• The limitations section is brief and does not mention potential issues such as recall bias in dietary data. While the recall period is 24 hours, it is still possible that mothers may forget some details. Additionally, the study's limitations should include a discussion on the interpretation and use of the MDD indicator. Although information is collected at the individual level, the indicator is more useful for population-level inferences and not for drawing individual-level conclusions. Dietary patterns can change overnight and may not reflect normal dietary habits in the household.

Minor Issues:

• Grammar, such as in the abstract: "variables associated to..." should be corrected to "associated with." Additionally, “Gahanna” in the results section should be corrected to “Ghana.” The use of lowercase letters to begin words in the tables (e.g., "primary education") should also be addressed.

• The reference style does not conform to the journal’s requirements.

Reviewer #2: The study aimed to assess the minimum dietary diversity and its determinants among lactating mothers in five Sub-Saharan African (SSA) countries (Kenya, Burkina Faso, Ghana, Mozambique and Tanzania). The manuscript gives an explanation of dietary diversity and has also highlighted the health and growth impacts of inadequate dietary diversity for both the mother and infant. The study setting is also very fitting because SSA is indeed facing several challenges particularly in relation to nutrition which is adversely impacting the health and general wellbeing of women and children especially. It is imperative for contextual research like this study, to shed more light on the issue and possibly generate data that can inform the development of targeted interventions. The study identified that dietary diversity among lactating mothers in the five sampled SSA countries was low due to an inter play of some individual and community level factors.

What I like about this study is the fact that, the authors highlighted the different and several nutritional related problems existing in Tanzania, Ghana, Mozambique, Burkina Faso and Kenya. Though having similar settings, there are yet some varied and unique features of each of these countries and the authors were able to pick out the very unique nutritional deficiencies and dietary inadequacies lactating mothers and their infants encounter in these different settings, providing us with a comprehensive understanding of the state of nutrition in SSA.

I also like the fact that the study did not only focus on the individual level factors as determinants of dietary diversity but delved into community level factors as well. Under the individual level factors, I was expecting to see variables like religion and ethnicity as these variables may impact behaviors, beliefs and practices that invariably influence dietary diversity. Since agriculture plays a role in food in security, it would have been nice to capture the food system including farming activities which may vary across the countries.

In all, the paper is good and I approve for it's acceptance and publication once the comments raised are addressed

Kindly find below additional comments to be addressed:

Major Points

1. Authors should clarify why sixteen food groups were used for assessing Minimum Dietary Diversity for Women (MDD-W). The food and Agricultural organization (FAO) recommend 10 food groups: i.e., grains, white roots and tubers and plantains, dark green leafy vegetables, meat, poultry and fish, pulses, nuts and seeds, vitamin A rich fruits and vegetables, other fruits, other vegetables, eggs and milk and milk products (FAO, 2021). Some food groups can be sub-divided to ensure clarity but at the point of calculating the dietary diversity scores, the sub groups are not included in the computation (FAO, 2021).

2. The recommended cut-off score by FAO is ≥ 5 food groups for adequate diversity and < 5 food groups for inadequate. Could Authors explain the choice cut off score used in the manuscript.

Minor points

1. Background, line 4, the word “of” has been omitted. Please correct.

2. Methodology, population and eligibility criteria: The age range mentioned there is “15-59 years”. The results section however indicated “15-49 years”. Please reconcile.

3. Methods: The “of” has been omitted from the first sentence “to assess the variation in minimum dietary diversity lactating mothers”

4. Under random effects in the methods please delete repeated word.

5. Double check and clarify the spelling “Gahanna”.

6. PLOS authors have the option to publish the peer review history of their article (what does this mean? ). If published, this will include your full peer review and any attached files.

**Do you want your identity to be public for this peer review?** For information about this choice, including consent withdrawal, please see our Privacy Policy .

Reviewer #1: **Yes: ** Dr Michael Boah

Reviewer #2: **Yes: ** Jemimah Atisepaa Akantoe

---

## [Author Response · Author response to Decision Letter 1]

26 Nov 2024

Response to reviewers has been uploaded.

---

## [Editor Report · Decision Letter 1]

3 Dec 2024

Minimum Dietary Diversity and its Determinants among Lactating Mothers in Five Sub-Saharan African countries: A multilevel analysis

PONE-D-24-31238R1

Dear Dr. Alebachew Ferede Zegeye,

We’re pleased to inform you that your manuscript has been judged scientifically suitable for publication and will be formally accepted for publication once it meets all outstanding technical requirements.

Kind regards,

Engelbert A. Nonterah, MD, PhD

Academic Editor

PLOS ONE
---

## [Editor Report · Acceptance letter]

PONE-D-24-31238R1

PLOS ONE

Dear Dr. Zegeye,

I'm pleased to inform you that your manuscript has been deemed suitable for publication in PLOS ONE. Congratulations! Your manuscript is now being handed over to our production team.

Kind regards,

on behalf of

Dr. Engelbert Adamwaba Nonterah

Academic Editor

PLOS ONE